# The Membrane Interactions of Nano-Silica and Its Potential Application in Animal Nutrition

**DOI:** 10.3390/ani9121041

**Published:** 2019-11-28

**Authors:** Marek Pieszka, Dorota Bederska-Łojewska, Paulina Szczurek, Magdalena Pieszka

**Affiliations:** 1Department of Animal Nutrition and Feed Sciences, National Research Institute of Animal Production, Krakowska Street 1, 32–083 Balice, Poland; dorota.bederska@izoo.krakow.pl (D.B.-Ł.); paulina.szczurek@izoo.krakow.pl (P.S.); 2Department of Animal Genetics, Breeding and Ethology, University of Agriculture, Mickiewicza Street 24/28, 30–059 Kraków, Poland; magdalena.pieszka@urk.edu.pl

**Keywords:** silica, nanoparticles, cell membranes, animal nutrition

## Abstract

**Simple Summary:**

Silicon dioxide nanostructures, due to good biocompatibility, low toxicity and high synthetic availability, are promising materials for various biological and industrial applications. Interest in using silicon dioxide nanostructures arises not only from their special interactions with cell membranes, but also from an ease in manipulating their particle size, shape and porosity, allowing one to make a material with the desired physicochemical properties. Despite that, there is still little known about the possible use of silicon dioxide and other nanostructures in animal nutrition. The aim of the present paper was to describe the properties of silica nanostructures, demonstrating potential applications and achievable benefits of using nanostructures as a feed additive. Based on the literature, it seems that diet supplementation with nanoparticles leads to improved performance and immunity in animals, which might be, at least partially, related to changes in the composition of gut microbiota. These unique features make nanoparticles interesting candidates as feed additives used in animal nutrition.

**Abstract:**

Nanoparticles are increasingly popular in numerous fields including electronics, optics and medicine (vaccines, tissue engineering, microsurgery, genomics and cancer therapies). The most widely used nanoparticles in biomedical applications are those designed by man. Scientists have obtained many types of silica nanoparticles with defined shape and chemical composition, but different properties and applications. Nanoparticles include particles with at least one dimension ranging from 1–100 nm. Silica nanoparticles (Sn), reaching values from several dozen to several hundred m^2^/g, have unique physicochemical properties due to their porous structure and well-developed specific surface. Currently, the use of Sn in animal nutrition, with a focus on gastrointestinal tract function, is of great interest.

## 1. Characteristics, Physicochemical Properties and Occurrence of Silica Nanostructures

Under natural conditions, silicon dioxide, or silica, is most often found in an amorphous form, e.g., as an opal or chalcedony, while at high temperatures in the interior of the Earth, it is present in crystalline forms such as quartz, tridymite or cristobalite. Silica is also formed as a result of slow decomposition of silicates under the influence of water and carbon dioxide [1,2,3]. Silicon dioxide obtained under laboratory conditions is a solid substance without an odor, occurring in the form of a white powder or colorless crystal. It is also a relatively inert, chemically passive substance, and, apart from an aqueous solution of hydrogen fluoride, it does not react with any other acid. However, the salts—silicates which have the ability to dissolve in water—are formed in the reaction with concentrated sodium or potassium hydroxide and their carbonates [1]. 

Silica nanoparticles (Sn), defined as nano-objects which three basic dimensions in the nanoscale, are characterized by optical transparency, low toxicity, and neutral pH, and many of them are also degradable. What is more, their surface can be more functional thanks to well-established silane technology. Silica nanoparticles have a very wide range of available sizes and can be synthesized as homogeneous particles from 5 to 2000 nm. They are most often synthesized using the Stöber method [4], i.e., they are formed in the process of hydrolysis and condensation of silicon alcohols, e.g., tetraethylorthosilicate (TEOS), in the presence of a catalyst—mineral acid (e.g., HCl) or base (e.g., ammonia) [4]. The simplest and most frequent method of nanoparticle surface functionalization is the use of silanes as anchor chemistry. There are many silanes available on the market that cover a wide range of functional groups that can be used for further reactions [5]. Mesoporous Sn is a large group, which is synthesized by the reaction of TFOS with a template made of micellar rods. The result is a collection of nano-dimensional balls or rods that are filled with a regular pore pattern. The template can then be removed by washing it with a solvent adjusted to the correct pH. According to IUPAC, mesoporous materials are defined as those that have a pore size in the range of 2–50 nm and an orderly pore system, giving it a regular structure [5]. The pore size can be varied and fine-tuned by selecting surface active agents such as MCM-41, MCM-48, SBA-15, and SBA-16, which are widely used to deliver drugs. In addition, they have also been studied as adsorbents, catalysts and biosensors.

The most interesting physical feature of Sn is their porous structure and well-developed specific surface area, reaching values from several dozen to several hundred m^2^/g, which determine the outstanding adsorption and ion exchange capacity [6,7]. The structure of porous Sn is presented in Figure 1. The surface of Sn consists of silanol (≡Si-OH) and siloxane groups (≡Si-O-Si≡). Silanol groups are the reactive centers that enable surface modification and demonstrate the hydrophilic nature of silica [1]. In addition, they give the Sn surface an acidic character due to the possibility of proton cleavage (≡Si-O- + H+). In turn, siloxane groups determine the hydrophobicity of the material. Thus, depending on the structure of the nanosilica surface and the quantitative advantage of the above-mentioned groups, it may have a hydrophilic or hydrophobic character [6].

Fumed silica (Fs), also known as pyrogenic silica because it is produced in flame, consists of microscopic droplets of amorphous silica agglomerated into branched chains with an average particle size of 15 nm. It has the form of a white powder with very low density but with a large specific surface area. Its three-dimensional structure causes an increase in viscosity, which makes it applicable in various industries as a thickener, carrier or filler [8]. Silica is used as an abrasive additive in toothpastes, an additive for powdered extinguishing media, and a sorbent of harmful organic compounds and heavy metal ions. It is also used in electrotechnology and electrochemistry as a filler for plastomers and elastomers, thermoplastic polyolefines, dispersion paint and varnish fillers, tabletting aids, silicone rubber hardeners, and fillers for chromatographic columns or biomedical sensors, e.g., in the detection of some cancers [9,10,11,12,13,14,15]. 

Research carried out on Aerosil^®^ preparations has shown that Fs is characterized by a surface area of up to 330 m^2^/g with a minimum density of 50 g/L. It contains more than 99.8% of silicon dioxide at a pH between 3.7 and 4.5. Its specific properties include control of rheology and thixotropy of liquids, binders and polymers, as well as moisture absorption and prevention of caking, which improves the flow ability of powders. It also prevents delamination of the substance and has a thickening effect. It protects against the formation of stains in paints. It is also added to synthetic rubber, silicone or synthetic resins as a substance, improving their mechanical strength [8]. It was also shown that the Fs used in Aerosil^®^ preparation has bactericidal properties against many pathogenic microorganisms, including *E. coli*, *Salmonella* or *Clostridium*, especially when modified with silver, gold and titanium ions, and is also used, inter alia, in dressings, feed additives or medicine vectors/carriers [16,17,18,19].

## 2. Silica Nanoparticle Interactions with Biomembranes

Cell function is intimately related to the presence of cell membranes. The eukaryotic cell membrane forms a flexible shell, made up of two layers of phospholipids and proteins. Its cross section size does not exceed 10 nm. The molecular composition of the lipid bilayer determines its physicochemical properties, while many essential cell functions are fulfilled by membrane proteins, including communication with the environment, transport of molecules, and receptor and enzymatic activities [21].

The bacterial cell envelope also consists of a cell wall, which in some species may be additionally covered with a peptidoglycan layer, which is the main difference between gram-positive and gram-negative bacteria [22]. It is generally known that cell viability depends on the size of the negative charge of its surface, which is determined by the presence of carboxyl groups in sialic acids in the outer membrane [23]. This negative charge may be affected by the positively charged ammonium groups of membrane phospholipids—lecithin and sphingomyelin. The size of the negative charge also depends on the number of molecules and ions adsorbed on the cell surface, and is highly significant for the cell interactions with charged microparticles, e.g., silica. It is assumed that the active adsorption of negatively charged Sn results from electrostatic interactions and occurs through the aforementioned ammonium groups of membrane phospholipids [24].

When discussing the results of the adsorptive interaction of the cell with Sn, it should be emphasized that in this case, the Sn is adsorbed onto the cell surface, and not vice versa. The average diameter of the majority of bacteria is in the range of 0.2–10 μm (*Staphylococcus* 0.5–1.5 μm, *Enterobacteriaceae* up to 2 μm) [25], while for comparison, the Sn diameter is 3–30 nm. The synergy of Aerosil^®^ particles towards erythrocytes was studied by Diociaiuti et al. [26] using electron microscopy complemented with hemolysis and radio wave dielectric spectroscopy to clarify the extent of morphological and functional modification induced by Aerosil^®^. It was stated that the effect depends on both the silica concentration and the incubation time. These results agreed with an interaction model based on membrane protein denaturation due to the electrostatic attraction between (SiO-) groups at the silica surface and proteins embedded in the membrane. The process is time-limited and reaches saturation after about 20 min. The extent of the damage is determined mainly by the ratio between cell and Aerosil^®^ surface, which is the Aerosil^®^ concentration. Lower damage was observed when less Aerosil^®^ surface per cell was available. In contrast, hard membrane damage was observed when the Aerosil^®^ surface was extensive. The experiments showed considerable membrane damage with small weight concentrations [26].

On the basis of the above results, a two-stage mechanism of interaction between Fs and the erythrocyte membrane appears. The first stage involves rapid adsorption of Fs particles with the cell surface, mainly due to electrostatic interactions, subsequent consolidation of existing contacts with hydrogen compounds and van der Wall’s forces, while denaturation of membrane proteins, the degree of which depends on the proportion of the size of the silica particles and protein molecules, is noted in the second stage. Particles smaller than 3–5 nm do not have a sufficient surface to break large protein molecules on it and therefore do not induce hemolysis. The contact plane of large particles as well as agglomerates of fine particles with the membrane is sufficient to break down protein molecules, which leads to agglutination and hemolysis.

The method of laser flow cytometry was used to test the membrane-like properties of Fs. This method is based on the detection of light-scattering and fluorescence signals coming out of the cell under the influence of a laser beam. As a result of a study by Gerashchenko et al. [27], it was found that intact erythrocytes are characterized by a more intense direct light-scattering compared to experimental ones. The inverse effect is observed when registering the side light-scattering (at a 90° angle); control erythrocytes disperse light less intensively than erythrocytes with SiO_2_ addition. It was found that Fs is characterized by more intense autofluorescence in the green spectrum range than erythrocytes. On the Gaussian histogram of the distribution of autofluorescence intensity, both emissions do not coincide and are widely separated. The form of the histogram of SiO_2_ particle fluorescence distribution clearly reflects heterogeneity in particle size (polydispersity). After addition of erythrocytes to the SiO_2_ suspension, a reduction in the fluorescence distribution was observed, in the area where they partially overlapped. The changes observed in the scattering of light affecting erythrocytes that were previously treated with Fs may be caused by cell membrane deformation [27]. Increasing the fluorescence intensity of erythrocytes in the Fs suspension may be explained as a result of adsorption of a significant number of fine fumed SiO_2_ particles with high autofluorescence by erythrocytes. During this time, the adsorption of membranes and the contents of lysed cells by silica particles leads to a damping of this particle fluorescence. Fluorescence analysis of the events with a large light-scattering signal showed that they are caused mainly by large SiO_2_ particles, associated with membranes before and after cell lysis. The above study also explored an effect on hemolysis of various concentrations of SiO_2_ suspensions: 0.001; 0.005; and 0.01%. At the same time, 0.005% SiO_2_ suspensions were fractionated to assess the relationship of hemolysis on particle size. The smallest particles were separated from the large ones by centrifugation with a force of 300 g for a period of 10 min and finally the suspension was partitioned into two equal volumes. After the addition of the erythrocyte suspension to the samples, the degree of hemolysis was determined according to the dynamics of the number of released hemoglobin molecules. Predictably, the dispersion with 0.01% SiO_2_, as well as the fraction with large particles, demonstrated an increased hemolytic activity. The most intense hemolysis occurred in the first minutes of contact, the hemolytic effect decreased with time, and it was inhibited in all tested samples after 90 min of incubation. The observed exponential decrease in hemolytic activity over time can be explained by the saturation of the particle surface with the components of the lysed cells. The possibility that the smallest particles absorbed on the membranes increase cell resistance to hemolysis was not excluded. The mechanism of this phenomenon may be caused by blocking the adsorptive centers of the membrane by small Fs, which inhibits the incorporation of large particles into the process [27]. It is expected that, as a result of interactions with silica particles, the lysis of erythrocytes is selective, depending on the density of the negative charge on the cell surface, which in turn is referred to as the “architecture” of the supermembrane matrix. This will cause decomposition of old erythrocytes that have smaller amounts of negatively charged silicic acid residues compared to the young cells. Thus, it is possible to select erythrocytes according to the size of the surface charge, which can be used to study the aging process.

Morphological and functional changes in erythrocyte membranes, caused by interaction with Fs, were examined by scanning electron microscopy and dielectric spectrophotometry. Suspensions with increasing concentration of silica (from 0.06 to 0.51 mg/mL) came into contact with the suspension of erythrocytes in isotonic buffer, after which changes in cell forms were dynamically observed for 10, 25 and 60 min. It was noticed that the membranes of erythrocytes, which came into contact with the sorbent, are deformed, with characteristic protrusions and appendages appearing on them. On this basis, it can be concluded that the erythrocytes and silica particles interact with each other in an obvious way. It is characteristic that the average volume of deformed erythrocytes does not change significantly. Determination of the weight of membrane proteins showed that it is significantly reduced in erythrocytes treated with Fs, which binds and discharges membrane proteins from the lipid layer via silicon groups. Observation of the membrane permeability showed that this factor also significantly decreases as a result of interaction with Fs. This confirms the assumption that the extraction or denaturation of integral membrane proteins is responsible for ion transport across the membrane under the influence of SiO_2_. A correlation between the ionic membrane permeability and the erythrocyte hemolysis degree was found. This correlation confirms that the basic link in the mechanism of silica particle adsorption on erythrocyte membranes is the denaturation of integral membrane proteins following an electrostatic interaction with negatively charged silica surface centers. It was also found that the degree of severity of all calculated changes in morphology and membrane functions correlates with both the prolongation of incubation time and an increase in silica concentration in the suspension [27].

The study was also carried out on the chemical properties and impact of bulls’ spermatozoa membrane structures with Sn, which led to their improved stabilization and was used to optimize the concentration of sperm in the portion of the semen aimed for insemination [28]. To accomplish this task, the spermatozoa suspension obtained from the bull sperm was treated with Sn (0.1–1.0%) and prepared for a period of 45 min. During this time, oligopeptides, the products of protein membrane hydrolysis, some of which contained carbohydrate components, passed into the solution. The concentration of selected components, including proteins, glycoproteins, hexoses and sialic acids, was determined and compared in the obtained hydrolyzate. It was observed that Sn addition to the cell suspension reduces the quantitative yield in the hydrolyzate of proteins and glycoproteins constituting sialic acids, which proves the participation of these structures in Sn binding. At the same time, the activity of hexose fragments was at the level of the control, which confirms the data obtained earlier on low hexose adsorption on silica.

The authors also attempted to explain the changes that take place in the lipid bilayer in the presence of Sn. For this purpose, spectrophotometric measurements of the sperm suspension were made for physiological pH values and at room temperature. It was noted that the luminescence spectrum of spermatozoa was typical for the luminescence spectrum of tryptophan-containing proteins found on the cell surface. The contact with Sn did not affect the spectrum image; however, the intensity of luminescence decreased slightly, which is related to the increase of light-scattering by the suspension. Measurements also involved the spectra of luminescence of sperm cells into which fluorescent probes, TNS (2-p-toluidinylnaphthalene-6-sulphonate), were introduced. It was found that when exposed to Sn, the membrane-associated TNS luminescence spectrum showed a slight infrared shift. This may indicate structural changes within the cell membranes. The results of the study of TNS luminescence bound to the sperm membrane fixation with iodine anions before and after the contact with Sn confirmed that the membrane components cause conformational changes under the influence of Sn [28]. Additionally, pyrene, which has the property of binding to non-polar regions of proteins and lipid bilayers, was used in order to determine the effect of Sn on the sperm surface charge in the presence of a fluorescent probe. It was established that the fixation constant of luminescence of pyrene exders with iodine anions decreases when the sperm cell is contacted with Sn. This is caused, as is known, by an increase in the negative charge of the cell surface [28].

An effect of silicas, whose surfaces were modified by numerous groups, e.g., ammonium and methyl, P and Al, on mast cells characterized by high sensitivity to foreign substances and their rapid feedback was investigated in the experiment of Marzaioli et al. [29]. The sample of silica at 2% of its concentration was introduced to a mast cell suspension obtained from the abdomen of rats, and incubated for 30 min, after which time the number of cells in the activation phase and the number of dead cells were counted. It turned out that the unmodified Sn triggered the highest number of active cells characterized by membrane deformation, formation of granules on the membrane and penetration into the cell, and ejection of the nucleus by the most loaded fragments of the membrane. Together with this, the number of dead cells after contact with unmodified SiO_2_ accounted for 23% of all cells, whereas silica with Al induced irreversible membrane violation and death of 62% of the cells. A high percentage of cell death was observed in the case of silica with phosphorus. Silica modified with methyl groups practically did not show any effects on the cells. The cytological examination carried out in this way showed that mast cells are sensitive to the hydrophilic dispersive Sn [29].

The main features of the mechanism of Sn interaction with cellular membranes can be illustrated as follows: the basis of interactions is the electrostatic attraction between the negatively charged surfaces of SiO_2_ particles and the positive charge of ammonium phospholipids constituting the membrane, i.e., lecithin and sphingomyelin. Concurrently, the formation of hydrogen compounds between silicon groups and amide groups of membrane proteins was observed. At the time when the fine particles block the adsorbent centers on the surface of the membrane, the large ones extract membrane proteins and phospholipids, destroying them. In this way, the Sn suspension exhibits a membrane-like effect with respect to erythrocytes, which manifests itself when the cell is in direct contact with silica particles.

Interesting data on Sn interactions with the cellular membrane receptor apparatus were obtained as a result of tracing the effect of sorbent on rosette-like lymphocytes. It was found that the sorbent binds a significant number of T-lymphocytes, and at the concentration of SiO_2_—6 mg/mL, their number (compared to the control) was almost half [30]. The number of B-lymphocytes also decreased, but to a lesser extent. It seems that this is due to the blocking of specific T-lymphocyte receptors by the sorbent, as the number of T-lymphocytes forming rosettes with four or more erythrocytes is significantly reduced. The number of T-lymphocytes forming rosettes with three erythrocytes in the control was 9.6 ± 2.2%; with four and more, it was 39.8 ± 5.2%. In the experimental group, these numbers were 11.2 ± 2.4% and 14.4 ± 4.3%, respectively. Sn practically does not affect the formation of B-lymphocyte rosettes [30].

Knowledge of the specificity of Sn interactions with cellular membranes is important in clinical practice from the point of view of choosing the right treatment for various pathologies, for example in the treatment of wounds (increase in the intensity of hemolysis). With a layer thickness of 2 nm and more, no hemolysis was observed. With a layer thickness of less than 1.5 nm, intensive hemolysis occurred. In this way, application of Sn preparations on the clean surface of the wound without exudate may lead to cell damage at the wound surface. Therefore, Sn application in the regeneration phase is pointless. It is justified to use Sn in inflammation, when it is absorbed on the surface of microorganisms during the formation of microbial conglomerates, blocking their active membrane fragments, adhesion factors and others, which results in their removal from the wound [31].

Similar living conditions are characteristic for the microflora of other ecosystems, inter alia, of the digestive tract of warm-blooded animals. Microorganisms turn out to be the basis of these systems, and their interaction with solid bodies is characterized by many factors, e.g., they are responsible for the mutual attraction of interacting surfaces, while others, inversely, ensure their repulsion [32,33]. The surface properties of solids have a significant influence on the metabolic activity of bacteria, which are mainly responsible for the sorbing forces and the availability of nutrient particles. Because the available surface plane increases with the reduction of particle size, a particularly significant effect on the physiological activity of microorganisms is observed when interacting with highly disperse materials, e.g., Sn [34]. It was shown that the surface charge is of high importance for the adsorptive interaction of cells with charged microparticles, e.g., silica [31,35,36]. The surface charge of the bacteria decreases when the pH of the environment is changed to acidic, and at certain pH values (at the isoelectric point), the electric charge of the microorganisms is zero. Further lowering the pH of the environment beyond the isoelectric point usually leads to charging the surface of microorganisms, which significantly contributes to the tethering of the amino groups therein [37].

Generalizing the data provided in the last chapter, it can be concluded that Sn interacts with functionally active fragments of the cell membrane, blocking them and thereby significantly changing the properties of the cells. Silica nanoparticles induce hemolysis of erythrocytes, which increases as the particle size increases and the concentration of silica in the aqueous suspension increases, therefore unlike carbon sorbents, most of which have a hem compatibility, the Sn suspension cannot be recommended for parenteral introduction.

## 3. The Potential Use of Nano-Silica in Animal Nutrition

The addition of nanoparticles to foods depends on the nature of their surface. Micelles, liposomes, nanoemulsions, and polymers are the most often used feed additives. In the food industry, nanotechnology is used mainly to reduce the use of preservatives [38]. Nanotechnology in livestock sectors seems to be very promising but still warrants further investigation. Studies have suggested that feeding with nanoparticles improves animals’ body weight gain, feed conversion ratio and immunity response. The positive effect of different nanostructures on intestinal pathogenic microflora makes them an interesting alternative to antibiotic growth promoters used in animal nutrition. Moreover, nano-additives are characterized by higher bioavailability and thus might be used in smaller doses, having a minimal impact on stability of other feed components. 

An improved body weight gain of pigs exposed to silver nanoparticles was observed in the study by Fondevila et al. [39]. It was noted that from the second week after weaning, there was a linear correlation between the doses of silver nanoparticles administrated (from 0.20 to 40 mg/kg) and the daily weight gain of the piglets. The observed effect could be related to changes in the composition of gut microbiota because, at the same time, a decrease in the number of bacteria from the coli group was also recorded. Similarly, a constant administration of silver nanostructures to rats (9, 18 or 36 mg/kg body weight, twice daily with an interval of 10 h) for 13 weeks resulted in a higher number of Gram-negative bacteria, which in turn may be associated with changes in the immune system response and its interaction with the intestine [16]. Interestingly, the effects of silver nanoparticles on intestinal microbes are different from those exerted by silver ions, and therefore their mechanism of action is also probably different or only partly caused by the release of these ions [40]. 

Another in vivo study conducted in mice also confirmed the modeling effect of silica and silver nanoparticles on intestinal microflora [41]. In this 28-day study performed in accordance with OECD Guideline 407, animals were orally treated with silica or silver nanoparticles at 5–500 ppm or 46–4600 pbb, respectively. The authors noted a dose-dependent reduction in the number of *Bacteroides* and a dose-dependent increase of *Firmicutes* in animals treated with silver nanoparticles, and a dose-dependent reduction in the *Actinobacteria* in mice treated with silica nanoparticles. Moreover, in a study conducted by Sawosz et al. [42], the administration of silver nanoparticles in drinking water to quails (*Coturnix japonica*) caused an increase in the content of lactic acid bacteria in the caecum, but only in the higher doses.

In our own study, we tested the impact of dietary supplementation with Aerosil^®^ 300 preparation (Evonik, Germany) containing Fs on piglets’ performance up to 63 days of age [43]. The study was carried out in 30 litters from sows of synthetic line 990, with about 10 piglets each. Aerosil^®^ 300 and pancreatic-like enzymes of microbial origin (protease and amylase, Amano Enzyme, USA) were added to piglets’ diet in the period between 7 and 13, as well as 14 and 20 days of age. The use of the additive containing pancreatic-like enzymes together with Sn significantly increased the body weight of piglets between 1 and 63 days of age. The best results were obtained in the group receiving the additive with Fs and amylase, and moreover these piglets were characterized by the lowest mortality rate and diarrhea prevalence in the post-weaning period [43]. 

Furthermore, studies carried out by Shi et al. [44] showed that supplementation with nano-selenium at 3.0 g/kg dietary dry matter in sheep decreased the ruminal pH (range of 6.68–6.80) and ammonia N concentration (range of 9.95–12.49 mg/100 mL), and increased the total VFA concentration (range of 73.63–77.72 mM). Another important observation was that the nano-selenium additive enhanced the testis Se content and the testicular and semen GSH-Px (Se-dependent glutathione peroxidase) activity, and protected the membrane system integrity and the tight arrayment of the midpiece of the mitochondria [45]. 

Wang et al. [46] noted that chromium nanocomposite inclusion at the rate of 200 μg in finishing pigs resulted in significant increments of plasma immunoglobulin M and G content. In this study, reduced serum levels of glucose, urea nitrogen, triglyceride, cholesterol and non-esterified fatty acid were also reported. Further research showed significantly increased serum levels of total protein, high density lipoprotein and lipase activity. Wang and Xu [47] used the same dose of chromium nanoparticles in finishing pigs and showed that the addition of nano-Cr significantly decreased the feed:gain ratio by 3.56%, and had beneficial effects on carcass characteristics, pork quality and individual skeletal muscle weight, as well as increased tissue chromium concentration in selected muscle and organs. Nanoparticles of chromium picolinate (Crpic) enhanced chromium digestibility and its absorption as showed by Lien et al. [48]. Chrome digestibility and serum Cr concentration were higher in rats fed nano-chromium picolinate than in rats fed chromium picolinate. Moreover, the average daily gain was significantly greater in rats fed nano-Crpic or Crpic than in control rats.

In turn, Gonzales-Eguia et al. [49] investigated the effects of nanocopper on copper availability, nutrient digestibility, growth performance and serum traits of piglets. The group of piglets supplemented with nanoCu showed statistically significantly improved growth performance, reduced copper excretion and improved copper availability. The size of the nanoparticles did not however influence the serum copper level and serum cholesterol concentrations, or the hematology traits. 

Similarly, supplementation of piglets diet with nanoZnO resulted in higher average daily weight gain [50], which was also shown in research carried out by Yang and Sun [51], where nanoparticles of ZnO lowered incidences of diarrhea in weaned piglets and improved their growth performance. Wang et al. [52] conducted a study to investigate the effects of dietary zinc oxide nanoparticles on growth, diarrhea rate and intestinal morphology in weaned piglets. The addition of 800 mg/kg nano-ZnOs and 3000 mg/kg ZnO significantly increased the average daily gain and decreased diarrhea occurrence. Moreover, 800 mg/kg nano-ZnO significantly reduced plasma diamine oxidase activity, decreased total aerobic bacterial population in mesenteric lymph nodes, enhanced mRNA expressions of occludin, ZO-1, IL-1β, IL-10, TNF-α, and ki67 in ileal mucosa, and increased villous height, width, crypt depth, and surface area. Li et al. [53] showed that dietary supplementation with nanoparticles of zinc oxide can increase zinc digestibility, serum growth hormone levels and enhance the immune response of weaning piglets. On the other hand, Milani et al. [54] did not find an effect of ZnO nanoparticles on pigs’ growth performance, except for the gain:feed ratio and diarrhea occurrence that improved linearly over days 1–7 of the experimental period. The effect of nano Zn as a feed additive was also tested in a study in cows, which led to improved immunological response and consequently reduced somatic cell count in subclinical mastitis [55]. In addition, an increased level in milk production was noted in the experimental animals. 

All above-mentioned results on the use of different nanostructures in animals’ diets are shown in Table 1.

## 4. Summary

The application of nanotechnology in animal nutrition includes the use of different nanoparticles as carriers of nutrients, drugs, probiotics, trace, micro and macro-elements and other active substances such as essential oils, aromas, antioxidants, vitamins, enzymes and polyphenols. It is expected that such nano-additives will have the advantage that they will be better absorbed, easier to use and stable in interactions with other ingredients, as well as able to effectively prevent oxidation and improve assimilability and bioavailability of carried substances. Nanoparticles in animal nutrition not only increase the bioavailability of the substances administered, but also reduce the amount of metabolites excreted, which will be a huge advantage from the point of view of environmental protection.

Despite the fact that silica nanoparticles are a relatively new material (first synthesized in 1992), the interest in their biological applications is huge. Their unique properties, such as regular particle size, uniform and regular pore size, large external surface, large pore volume, and high bioactivity (numerous functional groups on the internal and external surface), allow one to use them in numerous fields, including pharmacy, nanotechnology, medicine, chemistry and electrical engineering. Moreover, low toxicity, biocompatibility and selectivity of interactions, combined with their extraordinary biological properties, make silica nanoparticles an interesting candidate as a feed additive used in animal nutrition. The use of their unique properties such as inhibiting the development of pathogenic bacteria, removal of bacterial toxins and increasing digestive enzymes creates an alternative to the use of antibiotic growth promoters, acidifiers and enzymes. The possibility of using silica nanoparticles in animal nutrition as carriers of valuable biologically active substances should also be emphasized.

## Figures and Tables

**Figure 1 animals-09-01041-f001:**
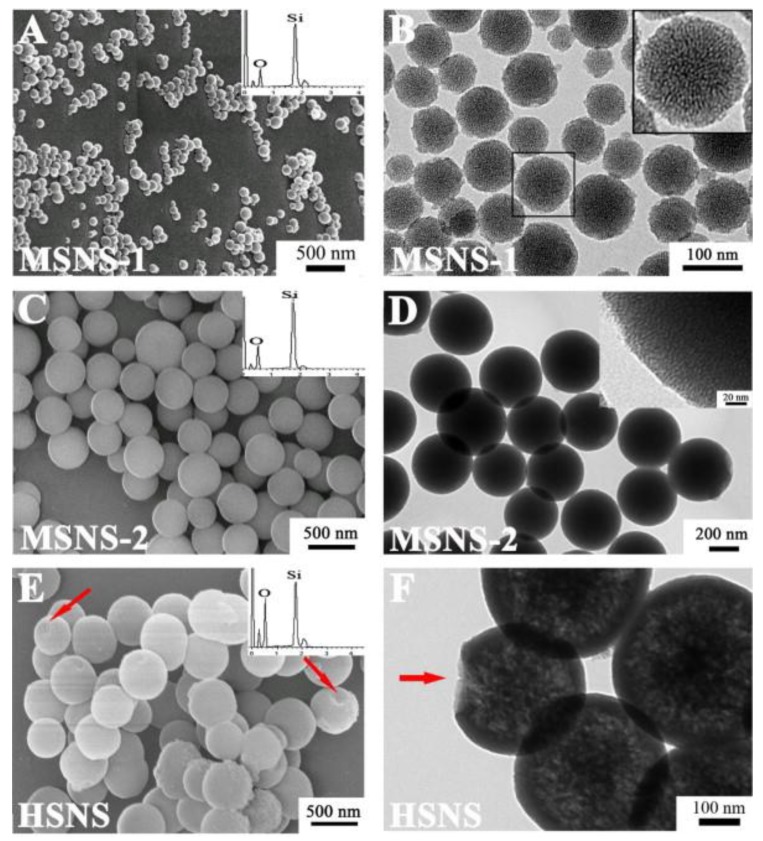
Scanning (**A**,**C**,**E**) and transmission electron microscopy (**B**,**D**,**F**) images of porous silica nanoparticles synthesized under different conditions (**A**,**B**) and (**C**,**D**), respectively) and porous hollow silica nanoparticles (**E**,**F**). Adopted from [20].

**Table 1 animals-09-01041-t001:** Results of studies on the efficacy of nanoparticle addition in animal nutrition.

Nanoparticles	Animal	Effect of the Experiment	References
Silver	Pigs	increased body weight gain antimicrobial effect in gastrointestinal tract	[39]
Silver	Rats	higher number of gram negative bacteria	[16]
Silver	Mice	increase number in *Firmicutes* bacteria decrease in *Bacteroidetes* population	[41]
Silica	Mice	reduced *Actinobacteria* population	[41]
Silver	*Coturnix japonica*	increased population of lactic acid bacteria	[42]
Silicon dioxide with pancreatic-like enzyme	Pigs	increased body weight gain lower mortality rate and diarrhea prevalence in the post-weaning period	[44]
Selenium	Sheep	decreased the ruminal pH and ammonia N concentration increased total VFA concentration	[45]
Selenium	Boer goats	enhanced the testis Se content, testicular and semen GSH-Px activity better protection of the membrane system integrity	[46]
Chromium	Pigs	increased level of plasma immunoglobulin M and G reduced serum levels of glucose, urea nitrogen, triglyceride, cholesterol and non-esterified fatty acid increased serum levels of total protein, high density lipoprotein and lipase activity	[47]
Chromium	Pigs	decreased feed:gain ratio beneficial effects on carcass characteristics increased in tissue chromium concentration in selected muscle and organs	[48]
Chromium picolinate	Rats	enhanced chromium digestibility and its absorption higher average daily weight gain	[49]
Copper	Piglets	better growth performance, reduced copper excretion and improved copper avability	[50]
Zinc oxide	Pigs	higher average daily weight gain	[51,52]
Zinc oxide		decreased occurrence of diarrhea	[51,52]
Zinc oxide	Pigs	better zinc digestibility, higher serum growth hormone levels and enhanced immune response	[53]
Zinc oxide	Pigs	improved gain: feed ratio decreased diarrhea occurrence	[54]
Zinc oxide	Cows	improved immunological response reduced somatic cell count in subclinical mastitis increased level in milk production	[54]

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
