# Peer review of "The Membrane Interactions of Nano-Silica and Its Potential Application in Animal Nutrition"

_animals, 2019, doi:10.3390/ani9121041_

Round 1
Reviewer 1 Report
I have read the previous version of the article. The current issue looks much better. Some of my suggestions have been addressed, but there is still some missing information that is necessary for such a review, and I cannot accept the item in the present form.
The review-type of article aims in the broad audience, and some BASIC knowledge about subjected materials should be included.
Note that nanostructured silica can have quite large scales – like porous silica thin layers, that can have 2D lengths equals several centimeters. Nanostructurization involves only some nanometer features in the structure. So distinguish nanostructured silica (in this case, cell penetration is not sure) and silica nanoparticles, that you are writing about.
In the title: change nanostructured silica to silica nanoparticles or nano-silica
I recommend supplementing work with some brief introduction to the nano-silica (nano-silica) since authors consider mainly nano-silica (silica particles with nanometer size). To be precise, describe (in brief) following issues:
1) definition of the silica nanoparticles (nano-silica)
2) types of nano-silica (solid, porous, SBA and MCM types and so on)
3) briefly describe methods of the synthesis (Stober, sol-gel…)
4) Figure 1 – wrong caption – it is only an example of MESOPOROUS silica, probably MCM-41 type. It some exception type of nano-silica (definitely not typical), so it should be clearly noted. I recommend to change this picture and show a few typical examples of various nanosilica (solid – irregular and spherical, porous spherical and SBA-15 for instance) and clearly explain differences.
If the authors supplement the article with suggested information I can accept the report.
Reviewer 2 Report
In this review article, the authors Marek Pieszka and Co have discussed the Membrane interactions of colloidal silica nanostructures and potential application of nanoparticles in animal nutrition. The revised manuscript has been improved now however, there are a few points that need to be addressed.
Authors have answered for comment 4 (Authors are suggested to include or discuss the application of silica nanoparticles in animal nutrition. Authors are suggested to include the following reference at Line No. 291 after feed components (Pathakoti et al., 2019). Pathakoti, K., et al., Nanoparticles and Its Potential Applications in the Agriculture, Biological Therapies, Food, Biomedical and Pharmaceutical Industries: A Review. 2019) as “According to the Reviewer suggestion we got familiar with proposed literature and we included it in our work” but have not found the reference in the manuscript.
Punctuation marks should be used at appropriate places in the article. Need to check for typographical errors, plagiarism, punctuation, and grammar throughout the manuscript.
Authors are suggested to include the future directions of membrane interactions of silica nanoparticles and potential application of nanoparticles in animal nutrition in summary.
Round 2
Reviewer 1 Report
All my remarks has been adressed correctly, I recomment the text for publication